# Differences in Penile Hemodynamic Profiles in Patients with Erectile Dysfunction and Anxiety

**DOI:** 10.3390/jcm10030402

**Published:** 2021-01-21

**Authors:** Rossella Cannarella, Aldo E. Calogero, Antonio Aversa, Rosita A. Condorelli, Sandro La Vignera

**Affiliations:** 1Department of Clinical and Experimental Medicine, University of Catania, 95123 Catania, Italy; rossella.cannarella@phd.unict.it (R.C.); acaloger@unict.it (A.E.C.); rosita.condorelli@unict.it (R.A.C.); sandrolavignera@unict.it (S.L.V.); 2Department of Experimental and Clinical Medicine, University Magna Graecia of Catanzaro, 88100 Catanzaro, Italy

**Keywords:** anxiety, erectile dysfunction, penile echo-color doppler ultrasound, GAD-7, IIEF-5, late-responder

## Abstract

Background: Penile echo-color Doppler ultrasound (PCDU) is the gold standard for the diagnosis of arterial erectile dysfunction (ED). Its reliability in patients with anxiety was questioned, due to false-positive results. Aim: To assess the penile hemodynamic response to intracavernous injection (ICI) of alprostadil in patients with anxiety-related ED. Methods: Patients with non-organic ED and a 5-item International Index of Erectile Function (IIEF-5) score ranging between 5 and 7 were enrolled. They were asked to compile the 7-item Generalized Anxiety Disorder (GAD-7) questionnaire to assess the degree of anxiety and were divided according to the GAD-7 score in Group 1 with minimal level of anxiety (*n* = 20), Group 2 with mild anxiety (*n* = 20), Group 3 with moderate anxiety (*n* = 20), and Group 4 with severe anxiety (*n* = 20). Peak systolic velocity (PSV) and the end-diastolic velocity (EDV) were sampled in all patients, through PCDU in the flaccid state, and 5, 10, 15, and 20 min after ICI of alprostadil at the standard dose of 10 μg. Results: In penile flaccidity, the patients showed a mean PSV of 8.0 ± 4.0 cm/s. The degree of anxiety was found to significantly influence both PSV and EDV at all assessed time-points. Particularly, it was negatively associated with the PSV at time 5 (*r* = −0.9, *p* < 0.01), 10 (*r* = −0.9, *p* < 0.01), 15 (*r* = −0.9, *p* < 0.01), and 20 (*r* = −0.7, *p* < 0.01) minutes, and positively with the EDV at time 5 (*r* = 0.7, *p* < 0.01), 10 (*r* = 0.6, *p* < 0.01), 15 (*r* = 0.5, *p* < 0.01), and 20 (*r* = 0.3, *p* < 0.01) minutes. Although all patients showed a mean dynamic PSV > 25 cm/s (which excluded an arterial ED according to the current guidelines), a peculiar hemodynamic pattern was found in patients with severe anxiety. In these patients, normal PSV values were reached only after 20 min from ICI, suggesting a “late-responder” profile. Conclusion: If further studies confirm the existence of a distinct hemodynamic profile in patients with severe anxiety, sampling the PSV and the EDV values could be proposed, for detecting patients with severe anxiety-related ED. Dynamic PCDU could be considered an accurate diagnostic test in patients with non-organic ED, since zero false-positive results were found in the present study. PSV in the flaccid state is not able to discriminate between arterial- or non-organic ED.

## 1. Introduction

Erectile dysfunction (ED) is defined as the persistent failure to achieve or to maintain a penile erection satisfactory for sexual performance [1]. Both organic and psychogenic causes, either alone or in combination, are recognized to play a role in its pathogenesis [2]. Penile erection results from a complex series of events requiring the integrity of the cavernous arteries and their smooth musculature, the tunica albuginea, the spongiosal, circumflex and cavernous veins, and the correct release, availability, and balance of neurotransmitters involved in relaxation of the arteries and arterioles’ smooth muscles [3]. In the penile flaccid state, cavernous smooth musculature is tonically contracted, allowing only a minimal arterial blood flow required for nutritional purposes. Sexual stimulation prompts the release of nitric oxide (NO), synthesized by the neuronal NO synthase (nNOS) enzyme via the parasympathetic nervous system. NO modulates the intracellular levels of cyclic GMP into the smooth muscle cells, in turn inducing vascular relaxation and, consequently, the increase of arterial blood flow, which leads to erection [3]. Anxiety associates with an exaggerated adrenergic tone. Cavernous muscle cells express α-adrenergic receptors and, once activated by the sympathetic neurotransmitters, they lead to arterial smooth muscle contraction and hinder penile erection. Several lines of evidence support the negative impact of anxiety on sexual performance and erectile function [4,5,6], and this negative influence is widely accepted [3,7].

Penile echo-color Doppler ultrasound (PCDU) examination after intracavernous injection (ICI) of prostaglandin (Pg) E1 or derivatives (e.g., alprostadil) is a second-line diagnostic test available for patients with ED. This test allows us to see single images, differently from the penile Duplex Doppler ultrasound (PDDU), where Color Doppler images are combined with the grayscale (B-mode) ones. This leads to the visualization of duplex ultrasonography images, needed for simultaneous visualization of the anatomy of the area [8]. Although due to the lack of standardization of the sampling location [9] and cut-off values, a relevant effort was made through decades to identify PCDU waveforms predictive of arterial or venous ED [3]. In greater detail, PCDU allows the measurement of peak systolic velocity (PSV) and end-diastolic velocity (EDV) in the cavernous artery, following the injection of alprostadil. Both of these parameters describe the blood flow characteristics in the cavernous artery during erection and are currently used for the diagnosis of vasculogenic ED. Accordingly, a restriction of the cavernous artery lumen (e.g., due to an arterial plaque or to a greater media-intima thickness) leads the arterial blood flow to slow down, and the PSV value decreases accordingly. In this regard, a PSV cut-off value ranging from 25 to 35 cm/s was suggested as the limit [10]. Conversely, in the case of venous ED (blood is drained too fast by the penile dorsal vein), EDV increases (>5 cm/s) [3].

Despite efforts to make the PCDU analysis an accurate method for the diagnosis of organic ED, scanty data are currently available on the PCDU outcome in anxiety-related ED. Anxiety is able to negatively impact on the quality of erection. Accordingly, a pioneer study described how psychogenic factors impact on erectile response to the intracavernous injection of papaverine [11]. Subsequently, another study described a lack of erectile response to ICI of alprostadil, at a dose of 20 μg, in patients with anxiety-related ED [12]. Anxiety-related ED affects young patients (≤30 years) more commonly than middle-aged ones [4]. However, how exactly anxiety influences the PCDU parameters and if a high state of anxiety associates with a specific hemodynamic response, is unknown. Therefore, we undertook a cross-sectional study to evaluate whether the PSV and the EDV before and after the ICI of alprostadil, are influenced by the degree of anxiety in patients with non-organic ED. The results of this study can help identify a typical anxiety-related penile hemodynamic pattern of response to alprostadil.

## 2. Patients and Methods

### 2.1. Ethical Aspects

A cross-sectional design using within-subject analysis was chosen as the experimental design for this study. It was performed in the Division of Endocrinology, Metabolic Diseases and Nutrition of the University Teaching Hospital “G. Rodolico”, University of Catania, Catania, Italy. The internal Institutional Review Board approved the protocol that was conducted according to the Good Clinical Practice. An exhaustive explanation of the study purpose was given to each participant and informed written consent was obtained in compliance with Helsinki’s declaration.

### 2.2. Patient Selection

Male patients referring to our Division for ED, older than 18 years, were considered for inclusion. Prior enrollment, all eligible patients underwent comprehensive medical history collection, physical examination, and hormone assessment. The 5-item International Index of Erectile Function (IIEF-5) questionnaire, which is a validated tool used to score the severity of ED, was administered to all patients. The 5-item questionnaire represents a short version of the IIEF-EF, and include only the erectile function domain. Scores range from 5 to 25. A score of 5–7 indicates a severe ED, a score of 8–11 a moderate one, 12–16 a mild-moderate ED, and 17–21 a mild form. The scores 22–25 indicate no ED. Only patients with an IIEF-5 score below 7, suggesting a severe ED, were recruited in this study [13].

The exclusion criteria were given by the presence of hypogonadism (total testosterone <264 ng/dL confirmed by at least two measurements [14]), hyperprolactinemia, hypothyroidism, diabetes mellitus, hypertension, dyslipidemia, overweight, and obesity; anamnestic factors or lifestyles potentially able to cause organic ED, cigarette smoke, urogenital surgery, and use of drugs for the treatment of chronic diseases in the three months before enrollment. Patients in mourning, and those who were divorced, or fired were also excluded as these might cause psychological or psychosexual disturbances that contribute to the pathogenesis of ED.

### 2.3. Study Protocol

Eighty patients satisfied the inclusion criteria. They had a mean age of 35.8 ± 8.6 years and were asked to complete the 7-item Generalized Anxiety Disorder (GAD-7) to assess the severity of anxiety. This is a validated self-administered 7-item questionnaire that investigates the occurrence of anxiety, lack of self-control, worrying, irritability, or fear in the 2 weeks before. Each question can be scored from 0 to 3 points, and overall, a GAD-7 score below 5 indicates minimal levels of anxiety, a score ranging between 5 and 9 indicates a mild anxiety, a score between 10 and 14 a moderate anxiety level and, finally, a score ranging from 15 to 21 a severe anxiety disorder [15]. The items of the GAD-7 questionnaire are showed in Appendix A.

According to the GAD-7 score, four groups were identified. Group 1, comprising patients with a GAD-7 score <5; Group 2, including patients with a GAD-7 score between 5 and 9; Group 3, comprising patients with a GAD-7 score between 10 and 14; and Group 4, including patients with a GAD-7 score ≥ 15. The study was designed to enroll a total of 80 patients, 20 for each group. Only patients with severe ED (IIEF-5 between 5 and 7) entered the study. These patients underwent basal and dynamic PCDU, as recommended by the ISSM Standards Committee [16]. In detail, dynamic PCDU for the evaluation of the cavernous arteries was carried out by a dedicated machine (Esaote MyLab 25, Genova, Italy) using broadband linear array transducer and color-power Doppler software, according to a procedure published earlier [10]. Flow parameters were blindly evaluated by trained examinators (S.L.V. and A.A.) and included PSVs and EDVs, before and after a pharmacostimulation test with a fixed dose of 10 mcg PGE1 at different time-points (5, 10, 15, 20 min). The resistance index (RI), calculated according to the following formula RI = (PSV-EDV/PSV), was used to confirm the diagnosis of venous ED, in case of EDV results higher than 5 cm/s.

### 2.4. Statistical Analysis

The patients were classified into four groups according to the GAD-7 score. Results were reported as mean ± standard error of the mean. The normal distribution of PSV and EDV was evaluated using the Shapiro-Wilk test. The age of patients and PSV values were normally distributed. EDV values were not normally distributed. Between-group PSV and EDV differences were analyzed by multivariate analysis of variance (MANOVA) and the post-hoc LSD test. To investigate the role of age and anxiety on PCDU results, age and GAD-7 scores were included in a multivariate regression analysis with the stepwise procedure performed for EDV and PSV, at each time of assessment. The chi-squared test was used to compare the percentage of patients with abnormal PSV and EDV among groups. To assess the relationship between PSV and EDV, a correlation analysis using the Pearson coefficient was used for each time-point. Statistical analysis was performed using SPSS 22.0 for Windows (SPSS Inc., Chicago, IL, USA). A p-value lower than 0.05 was accepted as statistically significant. A trend was assumed for p-values ranging from 0.05 to 0.099.

## 3. Results

At intergroup analysis, the age of groups 1 and 2 were significantly higher than that of Groups 3 and 4 (Table 1).

At MANOVA analysis, the degree of anxiety did not affect PSV in the flaccid state. It was 8.1 ± 2.0, below 13 cm/s in all patients. Between-group analysis showed a significantly negative effect of the degree of anxiety on dynamic PSV (Figure 1). In detail, Groups 1, 2, and 3 had significantly higher mean values than those of Group 4, after 5, 10, and 15 min. After 20 min, the PSV values of Groups 3 and 4 almost overlapped. At this time, Group 1 and 2 had a significantly higher PSV value than Group 4 (Appendix A). The effect size is shown in the Appendix A.

Notably, Groups 1, 2, and 3 had a higher PSV mean than the cut-off of 25 cm/s, at all investigated times. Interestingly, Group 4 showed abnormal PSV up to 15 min after ICI, but normalized after 20 min (Figure 1). This configured a hemodynamic pattern that could be defined as “late-responder”. Additionally, when a cut-off of 30 cm/s was considered, all patients of both Groups 1 and 2 reached a PSV ≥ 35 cm/s 20 min, after ICI. In contrast, only 70% and 75% of patients of Groups 3 and 4, respectively, reached this PSV cut-off value after 20 min from ICI (*p* < 0.01). The severity of anxiety correlated significantly and inversely with the PSV values. Particularly, a higher score was associated with a lower mean PSV, after 5 (*r* = −0.92, *p* < 0.01), 10 (*r* = −0.92, *p* < 0.01), 15 (*r* = −0.90, *p* < 0.01), and 20 (*r* = −0.74, *p* < 0.01) minutes from alprostadil ICI.

EDV values were significantly different among the groups at each time of evaluation (Appendix A). The effect size is shown in the Appendix A. In particular, a significantly higher mean EDV was observed in Group 4, as compared to the other groups, up to 10 min after ICI. EDV was <5 cm/s in all patients of all groups and at all times of investigation (Figure 2). Therefore, the IR was not calculated.

The severity of anxiety correlated significantly and positively with the EDV after 5 (*r* = 0.73, *p* < 0.01), 10 (*r* = 0.60, *p* < 0.01), 15 (*r* = 0.46, *p* < 0.01), and 20 (*r* = 0.26, *p* < 0.01) minutes. This indicates that the higher the GAD-7 score, the higher the EDV, with a strong association of up to 5 min, a moderate one at time 10, and 15 min, and a weak association 20 min after alprostadil ICI.

At the multivariate regression analysis, PSV and EDV were confirmed to significantly associate with GAD-7 score, independent of age, which was excluded from the model. Finally, although PSV and EDV measured different hemodynamic parameters, they were not completely independent. Indeed, the correlation analysis between these two hemodynamic parameters showed a strong correlation after 5 min from the injection of alprostadil and a weak or moderate one at later time-points (Appendix A).

## 4. Discussion

This study showed for the first time the different penile hemodynamic patterns in patients with non-organic ED, depending on the presence and degree of anxiety. We found that patients with a moderate or a severe anxiety disorder were younger than those with a minimal or a mild degree of anxiety. Furthermore, anxiety was found to influence the dynamic PCDU parameters after alprostadil ICI. The degree of anxiety correlated negatively with PSV and directly with dynamic EDV at all time-points of assessment. Interestingly, patients with the highest degree of anxiety (Group 4) had a distinct penile hemodynamic pattern, achieving normal PSV values only 20 min after alprostadil ICI. Similarly, this group of patients showed higher mean EDV values, after 5 and 10 min, as compared to the other groups. These ultrasound hemodynamic parameters confer the appearance of a “low-responder” pattern in patients with severe anxiety.

A previous study reported a significant association between ED and anxiety in young (≤30 years) patients but not in middle-aged (≥40 years) ones [4]. Hence, anxiety would more likely be involved in the pathogenesis of ED in the young, as compared to middle-aged patients [4]. A similar result was reported by Liao et al., who described a higher psychological distress in younger than in older patients with ED [6]. These findings are in agreement with those described in the present study, since patients with a lower degree of anxiety were older than those with a more severe anxiety disorder.

A clinical study recently reported a positive correlation between IIEF-5 and GAD-7 scores. A more severe ED was associated with a higher degree of anxiety [6], thus, further confirming the relationship between ED and anxiety. In the present study, we demonstrated that even penile hemodynamics is influenced by anxiety. Particularly, the higher the severity of the anxiety disorder, the lower was the mean dynamic PSV, at all times of observation. In addition, the higher the severity of the anxiety, the higher was the dynamic EDV value, with a stronger association in the first 5 min of assessment.

Dynamic PCDU examination is currently used in patients with ED and it is considered the gold standard for the diagnosis of arterial ED [10,17]. By comparing the penile angiography with dynamic PDCU in the same patients, a value of dynamic PSV below 25 cm/s was found to be associated with marked arterial damage with a high accuracy (both sensitivity and specificity ≥95%) [10,17]. Nevertheless, arterial sufficiency was defined for PSV values ≥35 cm/s [10]. The correct diagnosis of arterial ED was of remarkable importance for patient’s health, since, according to the “artery size hypothesis”, it could anticipate subsequent major cardiovascular events (MACE), by about three years [18]. Accordingly, diagnosis of ED was associated with a 25% greater risk of a subsequent MACE in a 9-year long prospective study [19].

The reliability of dynamic PCDU is questioned, due to the false-positive rate in patients with anxiety [20]. Indeed, an exaggerated sympathetic tone could interfere with perivascular smooth muscle cell relaxation, leading to abnormal PSV values, which might wrongly be addressed to primary arterial insufficiency, especially when re-dosing is not performed [21]. Of note, we found that all 80 patients with different degrees of anxiety had PSV values >25 cm/s after 20 min from alprostadil ICI at the dynamic PCDU. In patients with minimal and moderate anxiety, such values were >30 cm/s in all cases, and in the 70% and 75% of patients with moderate and severe anxiety, respectively. Hence, no false-positive cases were observed when the cut-off value of 25 cm/s was considered. Our results therefore suggest that false-positive findings could be avoided in patients with anxiety (also in those with a moderate or severe disorder). When a dose of 10 μg of alprostadil was given, the parameters were assessed at least up to 20 min following ICI, and the limit value of 25 cm/s was chosen. When a value of 30 cm/s was considered, the rate of false-positive was relatively low (20 or 25% in patients with moderate and severe anxiety, respectively).

Some effort was made to understand whether PSV measured in flaccid state might predict arterial ED. A cross-sectional study on 1346 patients reported that 13 cm/s was the most accurate PSV value in flaccidity to predict dynamic values <25 cm/s [22]. Interestingly, all patients enrolled in the present study showed flaccid PSV values below 13 cm/s, but none were diagnosed for arterial ED. Furthermore, PSV in flaccidity was not affected by the degree of anxiety. This indicated that PSV in the flaccid state was not sufficient to discriminate between patients with arterial ED and those with anxiety-related ED. This is rational if we consider that anxiety causes a scarce arterial flow, due to the lack of penile smooth muscle cell relaxation. As only patients with severe ED (IIEF-5 < 5) were included in the study, one concern might raise about the etiology of ED in those patients with mild or no anxiety, since organic causes of ED were ruled out. This point could be resolved considering that not all forms of non-organic ED are due to anxiety. Although patients in mourning, and those who are divorced or recently fired were not included in this study, the presence of other psychological factors leading to non-organic ED (e.g., depression, conflictual relationship with partner, extramarital sexual intercourse, etc.) could not be ruled out.

We are aware of some limitations of the present study. First of all, the determination of anxiety by a single, self-administered questionnaire does not allow precise identification of a psychiatric structured disturbance, in our cohort of patients. In greater detail, the lack of a deepened psychiatric counseling did not allow us to differentiate between an anxiety disorder and a state of anxiety. The GAD-7 score was developed to identify an anxiety disorder. Therefore, patients with higher GAD-7 scores had a greater chance to have an anxiety disorder than only an anxiety state. Second, we did not recommend audio-visual or manual self-stimulation of genitals after ICI testing, a procedure suggested by some authors to overcome high levels of anxiety that might be induced by the procedure [23]. However, we strongly believe that strict selection criteria for entering the study and the low heterogeneity of this cohort of patients might have limited the aforementioned biases.

## 5. Conclusions

In conclusion, we showed that anxiety is able to affect penile hemodynamics in patients with non-organic ED, depending on the degree of the anxiety disorder. Patients with severe anxiety show a “late-responder” penile hemodynamic pattern, with distinct PSV and EDV parameters, at all times of assessment. Furthermore, although PSV in flaccid state was below 13 cm/s in all patients, a dynamic PSV value >25 cm/s was found in all cases 20 min after ICI. Therefore, these results suggest that (i) if further studies confirm the existence of a distinct hemodynamic pattern in patients with severe anxiety, PSV and EDV measurements might be proposed to diagnose patients with severe anxiety-related ED; (ii) PCDU scan after alprostadil ICI could be considered an accurate diagnostic test in patients with non-organic ED, since we found zero false-positives when the evaluation was performed up to 20 min and a cut-off value of 25 cm/s was considered; and (iii) PSV in flaccid state was not able to diagnose between arterial and non-organic ED.

## Figures and Tables

**Figure 1 jcm-10-00402-f001:**
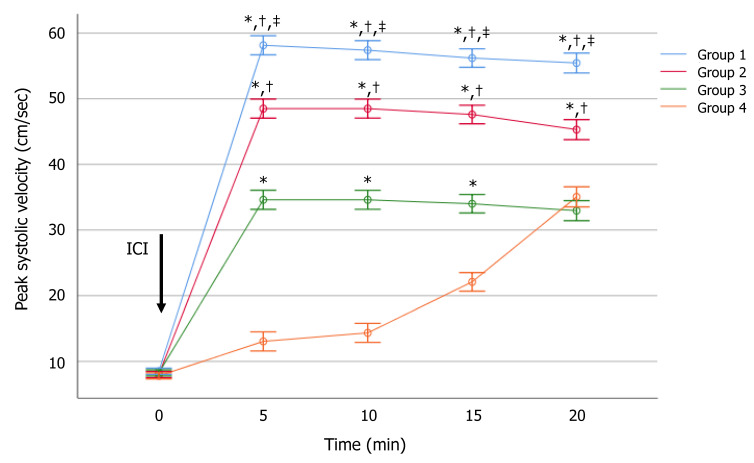
Effect of anxiety on the peak systolic velocity after intracavernous injection (ICI) of alprostadil at serial times of assessment. The level of anxiety was assessed using the Generalized Anxiety Disorder (GAD-7) score. Values are expressed as mean ± standard error of the mean. * *p* < 0.01 vs. Group 4; ^†^
*p* < 0.01 vs. Group 3; ^‡^
*p* < 0.01 vs. Group 2, analyzed using the multivariate analysis of variance (MANOVA), with LSD post-hoc analysis.

**Figure 2 jcm-10-00402-f002:**
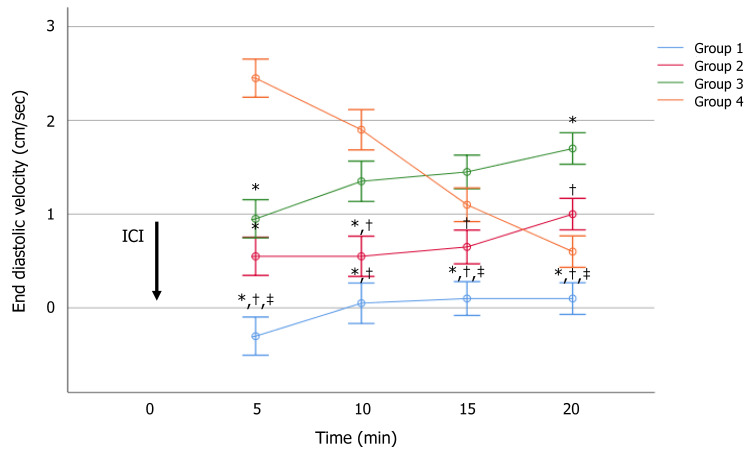
Effect of anxiety on the end-diastolic velocity after intracavernous injection (ICI) of alprostadil at serial times of assessment. The level of anxiety was assessed using the Generalized Anxiety Disorder (GAD-7) score. Values are expressed as mean ± standard error of the mean. * *p* < 0.05 vs. Group 4; ^†^
*p* < 0.05 vs. Group 3; ^‡^
*p* < 0.05 vs. Group 2, analyzed using the multivariate analysis of variance (MANOVA), with LSD post-hoc analysis.

**Table 1 jcm-10-00402-t001:** Age (years) of the patients.

Group 1 (*n* = 20)	Group 2 (*n* = 20)	Group 3 (*n* = 20)	Group 4 (*n* = 20)
39.20 ± 7.20 *^,†^	38.50 ± 7.70 *^,†^	30.70 ± 7.10	35.20 ± 9.90

* *p* < 0.01 vs. Group 3; † *p* < 0.01 vs. Group 4. The *p* was calculated by one-way ANOVA followed by the Duncan Multiple range test.

## Data Availability

The data presented in this study are available on request from the corresponding author. The data are not publicly available since they belong to an ongoing project.

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
