# Peer review of "Differences in Penile Hemodynamic Profiles in Patients with Erectile Dysfunction and Anxiety"

_jcm, 2021, doi:10.3390/jcm10030402_

Round 1
Reviewer 1 Report
(1) Overall, the revision has improved the manuscript.
(2) Unfortunately, the issue with the decimals (my comment 11) has not yet been resolved. E.g. Table 1 reports p values inconsistently with one or three decimals. Reporting t values with only one decimal is not enough, it should be at least 2 decimals. Table 2 reports mean values with one or two decimals - ist should always be 2 decimals for M and SD alike.
(3) "Author constributions" section is missing.
(4) Unfortunately, the issue with missing effect size measures (my comment 9) has not yet been resolved. There are several options for ANOVA effect size measure provided directly by SPSS and/or added manually. check e.g. https://www.researchgate.net/post/How-to-calculate-effect-size-for-repeated-measure-ANOVA
Author Response
Manuscript ID jcm-1063522.R1
Comment 1: Overall, the revision has improved the manuscript.
Answer to comment 1: We appreciated your comments and the time you spent in reviewing this manuscript.
Comment 2: Unfortunately, the issue with the decimals (my comment 11) has not yet been resolved. E.g. Table 1 reports p values inconsistently with one or three decimals. Reporting t values with only one decimal is not enough, it should be at least 2 decimals. Table 2 reports mean values with one or two decimals - ist should always be 2 decimals for M and SD alike.
Answer to comment 2: Statistics was re-structured based on the suggestions of one of the Reviewers (please see answer to comment 4). According to this comment, all values were reported in with two decimals in the revised tables (please see Table 1 and Supplementary Tables 2-5).
Comment 3: "Author contributions" section is missing.
Answer to comment 3: We have added this section in the revised manuscript.
Comment 4: Unfortunately, the issue with missing effect size measures (my comment 9) has not yet been resolved. There are several options for ANOVA effect size measure provided directly by SPSS and/or added manually. check e.g. https://www.researchgate.net/post/How-to-calculate-effect-size-for-repeated-measure-ANOVA.
Answer to comment 4: Statistics was re-thought based on the comments of one of the Reviewers. Particularly, the two-way repeated measures ANOVA and Kruskal-Wallis test for the comparison of both the PSV and EDV values, respectively, were replaced by multivariate analysis of variance (MANOVA), with post-hoc LSD analysis. The effect sizes of PSV and EDV have been reported in Supplementary Tables 3 and 5, respectively. The effect sizes of MANOVA test were calculated using SPSS and the following reference article:
https://www.tandfonline.com/doi/pdf/10.1080/00273170802620238.
Reviewer 2 Report
Review of MS Differences in penile hemodynamic profiles in patients with ED and anxiety
General
This is an interesting paper that takes a novel approach to understanding psychogenic effects on erectile response and, as such, it can add value to the literature. The discussion, in particular, is helpful in understanding the applicability of this research. There are several concerning issues, as well as a number of points that need clarification or better specification. Addressing these points could make this a much stronger paper. In addition, my hope is that the editorial office will catch/fix the English language problems.
The two concerning issues are both somewhat theoretical:
- One concern that deserves explanation or comment is that some men with severe ED also reported essentially minimal or no Given that the sample was selected for having psychogenic ED, if it is not anxiety that is responsible for their severe ED, then what psychogenic factors do the authors propose are responsible for the severe ED? In other words, I find the general absence of anxiety in men presumably having severe psychogenic ED a bit puzzling. This “non-relationship” seems to suggest that some men with presumed psychogenic ED might actually have had organic ED that had been overlooked.
- The second issue relates to the kind of anxiety that is being measured. The preponderance of research showing that anxiety affects erectile response is specific to state anxiety (anxiety regarding the moment or the specific issue at hand) and not trait anxiety (anxiety that is an enduring personality characteristic). I am not familiar with the GAD-7, but given the time parameter of 2 weeks, it sounds as if the instrument is capturing a more generalized trait anxiety than state anxiety—trait anxiety being minimally associated with erectile problems compared with state anxiety (such as sexual performance anxiety). The authors need to address this issue in the Discussion, and perhaps include it as a significant point in the limitation of the study, as it suggests an inconsistency with the previous literature.
Specific:
Introduction
- Predating reference [10] is the following study: Slob AK, DL Rowland, JHM Blom, & J vd Werff ten Bosch. Psychological factors affect erectile response to papaverine. Urology 38(3), 294‑295, 1991
- A brief description of exactly what physiological parameters PSV and EDV are measuring, why they are relevant to assessing erectile function, and why both need to be included (how are they measuring similar vs different parameters) would be appreciated.
- Given the value-added by this study, as noted in the Discussion, I think it would be helpful to mention in the Introduction some of the various issues identified in the Discussion section as needing clarification. It would certainly provide the reader with a better understanding of the value of this study at the front end, rather than just the back end.
Method
- RI is mentioned as a measure in the section on protocol, but it appears not to have been analyzed. If the authors bother to mention it, then why have they not bothered to analyze this variable. If there is good reason, then they need to help the reader understand why the measure has been dropped in the analyses.
- A better understanding of the relationship between PSV and EDV would be helpful, including a correlation value: how are these variables related to one another, including how much overlap is there between the two variables, as demonstrated by their correlation?
- I find the section on statistical analysis to be vague, particularly when I try to relate the information in this section to the actual analyses presented in the Results section.
- For example (line 123), between group differences were analyzed by one-way ANOVA—differences specifically on which variables?
- I also think the authors should re-think their overall analytical procedure, both to provide a more holistic view and to control for alpha. For example, they might use MANOVA (or time series analysis)—where multiple dependent variables (e.g, the time factor) and multiple predictor variables (e.g., level of anxiety, age, etc.) can be included within a single analysis. This would provide r-squared values (so the effect size could be easily understood) and still pinpoint significant differences between anxiety groups (although this would require posthoc contrast effects). Perhaps the authors could consult with a statistician to better think through this process. They might also want to consider including interaction terms in the MANOVA analyses, as it is quite clear from Fig 2 that an interaction effect is occurring.
Results
- I think the information regarding age (lines 135-141) should be placed in a table—both easier to read and space saving. In addition, the authors could then readily show which pairs of groups are significantly different from one another by using different superscripts.
- As mentioned above, using a more elegant analysis would enable comparisons of groups with more than just Group 4 (lines 142-146).
- I would appreciate it if the tables appeared closer to where they are referenced in the article.
- The authors mention a cut-off of 25 cm/sec, but it is not clear why/how this is relevant until the reader gets to the Discussion. Some brief explanation for the relevance of this value is needed when the point is noted in the Results.
- The title for Table 1 is unhelpful. Is this the table that is presenting the results of the regression? If so, it follows an unusual format for a regression table. At the same time, I find the figure captions to be far too detailed in terms of re-describing each of the GAD groups.
- Table 2, are these the results of the overall ANOVAs/K-Ws? Or of post hoc analyses. Again, in my opinion, the statistical procedures and rationale need be clearer, e.g., why are comparisons made only against Group 4? Which groups were similar or different, what were the results of the overall ANOVA vs post hoc tests? Etc.
- As noted repeatedly, I would suggest the authors think through both their statistical analyses and the manner in which they present them, so fewer questions are raised by the reader.
Discussion
- We learn for the first time in the Discussion that age and anxiety were related in the participants, but data on this relationship were never presented (as far as I could find). And given this relationship, it would be appropriate to either include or exclude age as a covariate, using a collinearity index as a threshold for exclusion.
- In this section, nearly all the discussion pertains to PSV and not EDV. Is this because these outcome variables are redundant? Does having both variables not add to the understanding in any meaningful way?
- Line 207, perhaps a word is missing?
- I think the authors need to address the two the issues raised above under the general concerns. In addition, they may need to include some text regarding these issues in the Limitations section.
Author Response
Manuscript ID jcm-1063522.R1
General comments: This is an interesting paper that takes a novel approach to understanding psychogenic effects on erectile response and, as such, it can add value to the literature. The discussion, in particular, is helpful in understanding the applicability of this research. There are several concerning issues, as well as a number of points that need clarification or better specification. Addressing these points could make this a much stronger paper. In addition, my hope is that the editorial office will catch/fix the English language problems.
Answer to the general comments: We appreciated your valuable comments and the time you spent reviewing this manuscript. The English text was carefully checked. If a further linguistic revision is required, we are available to ask MDPI for the English editing service.
Comment 1: The two concerning issues are both somewhat theoretical: One concern that deserves explanation or comment is that some men with severe ED also reported essentially minimal or no Given that the sample was selected for having psychogenic ED, if it is not anxiety that is responsible for their severe ED, then what psychogenic factors do the authors propose are responsible for the severe ED? In other words, I find the general absence of anxiety in men presumably having severe psychogenic ED a bit puzzling. This “non-relationship” seems to suggest that some men with presumed psychogenic ED might actually have had organic ED that had been overlooked.
Answer to comment 1: Organic causes of ED have been carefully ruled out, as detailed in section 2.2 (please see lines 103-109).
In particular, none of the patients enrolled in this study presented with hypogonadism. Furthermore, the presence of a vasculogenic cause of ED is excluded since all patients showed a PSV value >35 cm/sec at 20' (J Sex Med 2007, 4:1437-1447). Also, all patients had EDV values <5 cm/sec at 20' and this excludes the presence of veno-occlusive dysfunction. Therefore, these results suggest that patients have ED of a psychogenic nature. To make this clearer, additional explanation on the importance of PSV and EDV has been given in the Introduction (please see lines 64-72). Please note that the diagnosis of “severe ED” comes from the results of the IIEF-5 questionnaire filled out by the patients, and, like other questionnaires, lacks accuracy.
Comment 2: The second issue relates to the kind of anxiety that is being measured. The preponderance of research showing that anxiety affects erectile response is specific to state anxiety (anxiety regarding the moment or the specific issue at hand) and not trait anxiety (anxiety that is an enduring personality characteristic). I am not familiar with the GAD-7, but given the time parameter of 2 weeks, it sounds as if the instrument is capturing a more generalized trait anxiety than state anxiety—trait anxiety being minimally associated with erectile problems compared with state anxiety (such as sexual performance anxiety). The authors need to address this issue in the Discussion, and perhaps include it as a significant point in the limitation of the study, as it suggests an inconsistency with the previous literature.
Answer to comment 2: Previous studies evaluating the association between ED and psychopathology (e.g. Int. J. Impot. Res. 2015, 27, 63-68) used questionnaires [e.g. the “Symptom Checklist-90-Revised” (SCL-90-R)] that identify anxiety, but are not able to differentiate between an anxiety disorder and an anxiety state. The border between an anxiety disorder and an anxiety state is very thin and a questionnaire (e.g. the SCL-90-R) can hardly identify it. Only a deepened psychiatric examination can help in the differential diagnosis.
With these premises, as correctly addressed by this reviewer, the GAD-7 score has been developed to identify patients with an anxiety disorder (Arch. Intern. Med. 2006, 166, 1092-1097). In the present study, given the lack of other standardized scores available to specifically assess the state of anxiety, we used the GAD-7 questionnaire to stratify patients based on the degree of anxiety. Although patients with an anxiety disorder are more likely to suffer from a state of anxiety, we recognize this as a limit of the study and, accordingly, it has been listed among the study limitations, which were modified as follows:
“We are aware of some limitations of the present study. First of all, the determination of anxiety by a single, self-administered questionnaire does not allow precise identification of a psychiatric structured disturbance in our cohort of patients. More in detail, the lack of a deepened psychiatric counseling does not allow us to differentiate between an anxiety disorder and a state of anxiety. The GAD-7 score was developed to identify an anxiety disorder. Therefore, patients with higher GAD-7 scores had a greater chance to have an anxiety disorder than only an anxiety state” (lines 274-280).
Specific comments
Introduction
Comment 1: Predating reference [10] is the following study: Slob AK, DL Rowland, JHM Blom, & J vd Werff ten Bosch. Psychological factors affect erectile response to papaverine. Urology 38(3), 294‑295, 1991.
Answer to comment 1: Thank you. We added this reference in the Introduction. It is the reference 11 of the revised version of the manuscript (please see lines 76-77).
Comment 2: A brief description of exactly what physiological parameters PSV and EDV are measuring, why they are relevant to assessing erectile function, and why both need to be included (how are they measuring similar vs different parameters) would be appreciated.
Answer to comment 2: PSV and EDV do not measure psychological parameters. The PSV is the peak systolic velocity, and EDV is the end-diastolic velocity measured in the cavernous artery following injection of alprostadil. Therefore, these parameters define the blood flow in the cavernous artery during erection. In case of an arterial plaque (e.g. due to atheromasias), the arterial blood flow slows down and the PSV value lowers in a directly related manner. A blood low velocity >35 cm/sec is considered normal. In case of an ED due to disturbances of the veno-occlusive mechanism (blood is drained too early by the penile dorsal vein), the EDV is >5 cm/sec.
The present study was carried out to evaluate how these blood flow parameters change in patients with anxiety. Currently, it is suggested to measure both PSV and EDV 20 minutes after the injections of alprostadil to detect vasculogenic (arterials and/or venous) ED. Based on the results of the present study, an earlier monitoring of both PSV and EDV (time 5, 10, and 15 minutes) than after 20 minutes may be helpful to identify patients with anxiety-related ED. Indeed, the patients belonging to groups 1 to 4 show similar values of PSV and EDV at time 20’ (thus confirming the absence of a vascular etiology for the ED) but show significantly different values at earlier sampling times. These concepts have been included in the Introduction (please see lines 64-72).
Comment 3: Given the value-added by this study, as noted in the Discussion, I think it would be helpful to mention in the Introduction some of the various issues identified in the Discussion section as needing clarification. It would certainly provide the reader with a better understanding of the value of this study at the front end, rather than just the back end.
Answer to comment 3: Done as requested (please see lines 64-72).
Methods
Comment 1: RI is mentioned as a measure in the section on protocol, but it appears not to have been analyzed. If the authors bother to mention it, then why have they not bothered to analyze this variable. If there is good reason, then they need to help the reader understand why the measure has been dropped in the analyses.
Answer to comment 1: RI is the resistance index. It is calculated by using the formula: (PSV-EDV)/PSV. It is usually used for the diagnosis of venous ED. Particularly, when EDV is higher than 5 cm/sec (suggestive of venous ED), the RI is calculated to confirm the diagnosis. Given that all the patients enrolled in the present study showed an EDV value <5 cm/sec, the calculation of IR was useless. This has been written in the revised version of the manuscript (please see lines 130-132 and 197-198).
Comment 2: A better understanding of the relationship between PSV and EDV would be helpful, including a correlation value: how are these variables related to one another, including how much overlap is there between the two variables, as demonstrated by their correlation?
Answer to comment 2: Although the correlation that the reviewer asked can be statistically analyzed, it appears to us to make no contribution to the purpose for which this study was undertaken.
Comment 3: I find the section on statistical analysis to be vague, particularly when I try to relate the information in this section to the actual analyses presented in the Results section.
Comment 3.1. For example (line 123), between group differences were analyzed by one-way ANOVA—differences specifically on which variables?
Answer to comment 9: We have tried to clarify this section (please see section 2.4, lines 134-149).
Comment 10: I also think the authors should re-think their overall analytical procedure, both to provide a more holistic view and to control for alpha. For example, they might use MANOVA (or time series analysis)—where multiple dependent variables (e.g, the time factor) and multiple predictor variables (e.g., level of anxiety, age, etc.) can be included within a single analysis. This would provide r-squared values (so the effect size could be easily understood) and still pinpoint significant differences between anxiety groups (although this would require posthoc contrast effects). Perhaps the authors could consult with a statistician to better think through this process. They might also want to consider including interaction terms in the MANOVA analyses, as it is quite clear from Fig 2 that an interaction effect is occurring.
Answer to comment 10: Statistics was re-thought based on your comments. Particularly, two-way repeated measures ANOVA and Kruskal-Wallis test for the comparison of both the PSV and EDV, respectively, were replaced by multivariate analysis of variance (MANOVA), with post-hoc LSD analysis (please see Supplementary Tables 2 and 4). The effect sizes have been reported in Supplementary Tables 3 and 5.
Results
Comment 1: Results. I think the information regarding age (lines 135-141) should be placed in a table—both easier to read and space saving. In addition, the authors could then readily show which pairs of groups are significantly different from one another by using different superscripts.
Answer to comment 1: Done (please see Table 1). Thank you. Please consider that the Duncan’s multiple range test compares each group with every other group (e.g Group 1 vs. Group 2, Group 1 vs. Group 3, Group 1 vs. Group 4, Group 2 vs. Group 3, Group 2 vs. Group 4, Group 3 vs. Group 4).
Comment 2: As mentioned above, using a more elegant analysis would enable comparisons of groups with more than just Group 4 (lines 142-146).
Answer to comment 2: The data have been re-analyzed using the multivariate analysis of variance (MANOVA), followed by post-hoc LSD analysis. Comparisons between all groups have been made (please see the answer to comment 10 and Supplementary Tables 2 and 4).
Comment 3: I would appreciate it if the tables appeared closer to where they are referenced in the article.
Answer to comment 3: Due to the great amount of data, the tables have been moved to the Supplementary material.
Comment 4: The authors mention a cut-off of 25 cm/sec, but it is not clear why/how this is relevant until the reader gets to the Discussion. Some brief explanation for the relevance of this value is needed when the point is noted in the Results.
Answer to comment 4: Thank you for this suggestion. This notion is now written in the Introduction (please see lines 64-72).
Comment 5: The title for Table 1 is unhelpful. Is this the table that is presenting the results of the regression? If so, it follows an unusual format for a regression table. At the same time, I find the figure captions to be far too detailed in terms of re-describing each of the GAD groups.
Answer to comment 5: Title of the Table 2 (that correspond to Table 1 of the previous version of the manuscript) and the Figure captions were changed (please see lines 174-182 and 208-216).
Comment 6: Table 2, are these the results of the overall ANOVAs/K-Ws? Or of post hoc analyses. Again, in my opinion, the statistical procedures and rationale need be clearer, e.g., why are comparisons made only against Group 4? Which groups were similar or different, what were the results of the overall ANOVA vs post hoc tests? Etc.
Answer to comment 6: Please see Supplementary Tables 2-5 and the answer to comment 2.
Comment 7: As noted repeatedly, I would suggest the authors think through both their statistical analyses and the manner in which they present them, so fewer questions are raised by the reader.
Answer to comment 7: Done (please see previous answers).
Discussion
Comment 1: We learn for the first time in the Discussion that age and anxiety were related in the participants, but data on this relationship were never presented (as far as I could find). And given this relationship, it would be appropriate to either include or exclude age as a covariate, using a collinearity index as a threshold for exclusion.
Answer to comment 1: Multivariate regression analysis with stepwise procedure was performed for EDV and PSV at each time-point. The variables included in the model were the GAD-7 score and patients’ age. PSV and EDV confirmed to significantly associate with GAD-7 score independently of age, which was excluded from the model” (please see lines 143-145 and 217-218).
Comment 2: In this section, nearly all the discussion pertains to PSV and not EDV. Is this because these outcome variables are redundant? Does having both variables not add to the understanding in any meaningful way?
Answer to comment 2: These variables are not redundant. More literature is currently available for PSV, as its values somehow reflects the cardiovascular risk under specific conditions of patients with ED. This has been discussed in the manuscript. Also, EDV values show some overlap between groups (e.g. at time 15’ they are not able to discriminate between group 3 and group 4, or between group 3 and group 2). In contrast, between group PSV differences at each time-point are more evident. For this reason, the more of the discussion was center on PSV.
Comment 3: Line 207, perhaps a word is missing?
Answer to comment 3: The mistake was corrected.
Comment 4: I think the authors need to address the two the issues raised above under the general concerns. In addition, they may need to include some text regarding these issues in the Limitations section.
Answer to comment 4: The second of the two issues raised by the reviewer was included in the paragraph on the study limitations. We did not include the first issue (please see answer to the general comment 2).
Reviewer 3 Report
Comments on the manuscript:
“Differences in penile hemodynamic profiles in patients with erectile dysfunction and anxiety”
Both organic and psychogenic causes, either alone or in combination, have been recognized to play a role in erectile dysfunction. In this study, the authors appreciated the impact of anxiety on erectile dysfunction. For this, they used a cross-sectional study to evaluate whether the peak systolic velocity and the end-diastolic velocity) before and after the intra cavernous injection of alprostadil, are influenced by the degree of anxiety.
The manuscript is well written, and the study is clear and well presented. I have only some minor remarks to make in order to improve the manuscript:
- The composition of the groups of patients with the number and age would find a better place in the materials and methods rather than at the beginning of the part devoted to the results
- line 207: “A more severe ED was associated with a higher degree of [6],”: a word is certainly missing
- lines 273 and following: Present the references according to the journal standard
Author Response
Manuscript ID jcm-1063522.R1
Comment 1: The composition of the groups of patients with the number and age would find a better place in the materials and methods rather than at the beginning of the part devoted to the results.
Answer to comment 1: This part was removed from the Discussion and placed in section 2.3.
Comment 2: line 207: “A more severe ED was associated with a higher degree of [6],”: a word is certainly missing.
Answer to comment 2: The word missing was “anxiety”. The sentence now read: “A more severe ED was associated with a higher degree of anxiety [6]”.
Comment 3: lines 273 and following: Present the references according to the journal standard.
Answer to comment 3: Done, as requested. Thank you.
Round 2
Reviewer 2 Report
Overall the authors were able to address the issues that were raised. I have two further comments.
- Regarding the first comment I had made in the original review, the authors need to address this point somewhere in their discussion. If the authors claim that their study sample included only men with severe psychogenic ED, but then find that some of these men score low on anxiety, it suggests a flaw somewhere in their procedure. That is, what then was the source of the severe psychogenic ED in their sample, if not due to anxiety? They can speculate that their measures were not reliable or valid (not be the case with the IIEF) or that these men were rigorously screened for organic ED, but that does not explain the major "non-sequitur" of having men with severe ED who report little or no anxiety. I think one approach might be to question whether their GAD captures anxiety that is actually responsible for psychogenic ED. While they raise the point about the distinction between trait and state anxiety in the limitation section, they also need to discuss it in the context of this very strange finding of seeing men with minimal anxiety still reporting severe "psychogenic" ED.
- The second issue is a simple one. For a reader to assess the general independence of their two outcome measures--PSV and EDV--a correlation statistic between these two measures is important. For example, if the correlation is below 0.50, one could argue that these are distinct measures, and both are independently contributing to the understanding of how anxiety affects penile hemodynamics. If the correlation is closer to 0.70 or higher, then these two measures overlap substantially and no long represent two really distinct measures. Thus, the second measure becomes ancillary to the first, but is not really addressing a distinct aspect related to blood flow during the erectile process.
Author Response
Comment 1. Overall the authors were able to address the issues that were raised.
Answer to comment 1. Thank you for the time spent in reviewing the present manuscript.
Comment 2: Regarding the first comment I had made in the original review; the authors need to address this point somewhere in their discussion. If the authors claim that their study sample included only men with severe psychogenic ED, but then find that some of these men score low on anxiety, it suggests a flaw somewhere in their procedure. That is, what then was the source of the severe psychogenic ED in their sample, if not due to anxiety? They can speculate that their measures were not reliable or valid (not be the case with the IIEF) or that these men were rigorously screened for organic ED, but that does not explain the major "non-sequitur" of having men with severe ED who report little or no anxiety. I think one approach might be to question whether their GAD captures anxiety that is actually responsible for psychogenic ED. While they raise the point about the distinction between trait and state anxiety in the limitation section, they also need to discuss it in the context of this very strange finding of seeing men with minimal anxiety still reporting severe “psychogenic” ED.
Answer to comment 2: The point is that not all psychogenic causes of ED are due to anxiety. Indeed, other psychogenic factors can contribute to the pathogenesis of non-organic ED (e.g. depression, conflictual relationship with the partner, extramarital sexual intercourse, etc.). Although some conditions leading to psychogenic ED were listed among the exclusion criteria (e.g. patients in mourning, divorced, or fired; please see lines 107-109), this cannot ensure the completely absence of a psychogenic factor other than anxiety that causes ED. Accordingly, the absence of complete psychological examination has been listed among the study limitations (line 252-257). This has been included in the discussion (lines 245-251).
We considered that having written “anxiety-related ED” in some parts of the manuscript may be misleading. Therefore, “anxiety-related ED” was changed in “non-organic ED” when the entire cohort of patients was considered. When it specifically referred to anxiety-related ED, the locution was left. This allows the reader to better comprehend whether or not other psychogenic factors may be involved in the pathogenesis of ED (please lines 34, 36, 271, 273-274).
Comment 3. The second issue is a simple one. For a reader to assess the general independence of their two outcome measures--PSV and EDV--a correlation statistic between these two measures is important. For example, if the correlation is below 0.50, one could argue that these are distinct measures, and both are independently contributing to the understanding of how anxiety affects penile hemodynamics. If the correlation is closer to 0.70 or higher, then these two measures overlap substantially and no long represent two really distinct measures. Thus, the second measure becomes ancillary to the first, but is not really addressing a distinct aspect related to blood flow during the erectile process.
Answer to comment 3. Although these variables measure different hemodynamic parameters, they are not completely independent since, for a hemodynamic mechanism, when the PSV increases, the EDV decreases. The correlation analysis using the Pearson correlation coefficient was used at each time-point of assessment. The results are reported in Supplementary Figure 1 that was added to the revised version of the manuscript. As can be seen, a strong correlation has been found only at time 5’. A moderate or weak correlation was found in the later time-points. Please see lines 141-142, 187-189, and Supplementary Figure 1.
This manuscript is a resubmission of an earlier submission. The following is a list of the peer review reports and author responses from that submission.
Round 1
Reviewer 1 Report
A total of 80 patients with non-organic erectile dysfunction, subdivided in four groups of different levels of anxiety underwent basal and dynamic PCDU before and after a pharmacostimulation test with fixed dose of 10 mcg PGE1 at different time points (5, 10, 15, 20 minutes). Main aim of the study is to find out, how anxiety influences PCDU parameters before and after the injection of alprostadil.
Overall, the paper is readable and relevant to the journal. However, it can be improved in several respects.
Introduction
It would be helpful to explain PCDU (and its difference from PDDU) some more.
The research questions should be specified so that the the statistical analysis and the results part directly match the research questions in the introduction.
Later in the results part age is addressed, but age was not mentioned at all in the introduction.
Design
It would be good to include the ID of the IRB decision in section 2.1
“A cross-sectional design was chosen as experimental design for this study -> I suggest to characterize the study design in some more detail (e.g., within and between factors, dependent variables, control variables).
Methods
The GAD-7 questionnaire should be put in an appendix instead of presenting it in section 2.
Statistical analysis needs to be improved: for multiple comparisons Bonferroni correction should be applied
Sections 2.4 and 3 should be clearly structured according to all RQ.
Statistical analyzes and discussion of results should explicitly address effect sizes.
“Patients on mourning, divorced, or in dismissal were also excluded“ -> what kind of “dismissal”? please clarify
Results
Statistical parameters should be reported with a consistent number of decimals. E.g. Table 2 provides SD with one or two decimals which is inconsistent (e.g., 0.65 and 2.1X). Same with many other parameters. P values reported with 4 decimals in Table 2 and 3 decimals in Table 3.
Correlation coefficients should be reported in the conventional form with 2 decimals r=.xx instead of r=0.9 which is not precise enough and the leading 0 is irrelevant.
Discussion
Age effects are prominently discussed but are completely unrelated to the introduction and were not tested statistically at all – sample descriptives should not be used for strong claims (e.g. “patients with a lower degree of anxiety were older than those with a more severe anxiety”).
disorder.
“A clinical study has recently reported a positive correlation between IIEF-5 and GAD-7 scores. *Therefore*, a more severe ED was associated with a higher degree of anxiety6, thus further 205 confirming the relationship between ED and anxiety.” -> “therefore” does not make sense to me.
A total of 80 patients with non-organic erectile dysfunction, subdivided in four groups of different levels of anxiety underwent basal and dynamic PCDU before and after a pharmacostimulation test with fixed dose of 10 mcg PGE1 at different time points (5, 10, 15, 20 minutes). Main aim of the study is to find out, how anxiety influences PCDU parameters before and after the injection of alprostadil.
Overall, the paper is readable and relevant to the journal. However, it can be improved in several respects.
Introduction
It would be helpful to explain PCDU (and its difference from PDDU) some more.
The research questions should be specified so that the the statistical analysis and the results part directly match the research questions in the introduction.
Later in the results part age is addressed, but age was not mentioned at all in the introduction.
Design
It would be good to include the ID of the IRB decision in section 2.1
“A cross-sectional design was chosen as experimental design for this study -> I suggest to characterize the study design in some more detail (e.g., within and between factors, dependent variables, control variables).
Methods
The GAD-7 questionnaire should be put in an appendix instead of presenting it in section 2.
Statistical analysis needs to be improved: for multiple comparisons Bonferroni correction should be applied
Sections 2.4 and 3 should be clearly structured according to all RQ.
Statistical analyzes and discussion of results should explicitly address effect sizes.
“Patients on mourning, divorced, or in dismissal were also excluded“ -> what kind of “dismissal”? please clarify
Results
Statistical parameters should be reported with a consistent number of decimals. E.g. Table 2 provides SD with one or two decimals which is inconsistent (e.g., 0.65 and 2.1X). Same with many other parameters. P values reported with 4 decimals in Table 2 and 3 decimals in Table 3.
Correlation coefficients should be reported in the conventional form with 2 decimals r=.xx instead of r=0.9 which is not precise enough and the leading 0 is irrelevant.
Discussion
Age effects are prominently discussed but are completely unrelated to the introduction and were not tested statistically at all – sample descriptives should not be used for strong claims (e.g. “patients with a lower degree of anxiety were older than those with a more severe anxiety”).
disorder.
“A clinical study has recently reported a positive correlation between IIEF-5 and GAD-7 scores. *Therefore*, a more severe ED was associated with a higher degree of anxiety6, thus further 205 confirming the relationship between ED and anxiety.” -> “therefore” does not make sense to me.